A hybrid attention network with convolutional neural network and transformer for underwater image restoration

Jiao Zhan 1
Wang Ruizi 1
Zhang Xiangyi 2
Fu Bo fubo@lnnu.edu.cn 2
Thanh Dang Ngoc Hoang thanhdnh@ueh.edu.vn 3
1 Liaoning Vocational College of Light Industry , Dalian , Liaoning , China
2 Liaoning Normal University , Dalian , Liaoning , China
3 Department of Information Technology, College of Technology and Design, University of Economics Ho Chi Minh City , Ho Chi Minh City , Vietnam
Angiulli Giovanni
Electronic publication date: 2023 Nov 7
Publication date: 2023
Volume: 9
Electronic Location ID: e1559
Received 2023 May 25; Accepted 2023 Aug 9
Copyright: ©2023 Jiao et al.
Copyright year: 2023
Copyright holder: Jiao et al.
License: This is an open access article distributed under the terms of the Creative Commons Attribution License, which permits unrestricted use, distribution, reproduction and adaptation in any medium and for any purpose provided that it is properly attributed. For attribution, the original author(s), title, publication source (PeerJ Computer Science) and either DOI or URL of the article must be cited.
License URL: https://creativecommons.org/licenses/by/4.0/

Keywords: Underwater image restoration, Transformer, Convolutional neural network, Hybrid attention, Attention mechanism, Image processing, Deep learning

Funding: General project of Liaoning Provincial Department of Education, China LJKZ0986 Postdoctoral Science Foundation University of Economics Ho Chi Minh City (UEH), Ho Chi Minh City, Vietnam This study was funded by the General project of Liaoning Provincial Department of Education, China, No. LJKZ0986; Postdoctoral Science Foundation, to Bo Fu. This research was funded by University of Economics Ho Chi Minh City (UEH), Ho Chi Minh City, Vietnam, to Dang Ngoc Hoang Thanh. The funders had no role in study design, data collection and analysis, decision to publish, or preparation of the manuscript.

==============================
The analysis and communication of underwater images are often impeded by various elements such as blur, color cast, and noise. Existing restoration methods only address specific degradation factors and struggle with complex degraded images. Furthermore, traditional convolutional neural network (CNN) based approaches may only restore local color while ignoring global features. The proposed hybrid attention network combining CNN and Transformer focuses on addressing these issues. CNN captures local features and the Transformer uses multi-head self-attention to model global relationships. The network also incorporates degraded channel attention and supervised attention mechanisms to refine relevant features and correlations. The proposed method fared better than existing methods in a variety of qualitative criteria when evaluated against the public EUVP dataset of underwater images.

Introduction

With the progressive acceleration of marine resource exploration and study as well as the promotion of strategic analysis of ocean development, humanity has just entered a phase of extensive ocean development and use. In the fields of marine energy exploration, marine environmental protection, and marine species analysis, underwater images contain a large amount of visual information about marine resources and are an important carrier for people to observe and explore the ocean. Real-world underwater images are frequently challenging to capture due to the dynamic underwater environment and the presence of many interference factors like light absorption. In addition, diffraction, polarization, absorption, scattering, color loss, and light attenuation also frequently appear (Raihan, Abas & De Silva, 2019). The process of underwater image acquisition usually leads to the degradation of image visual quality, such as image blur, color cast, noise, and so on, which will seriously affect the subsequent image analysis tasks, and it is impossible to obtain effective and accurate image visual content. Restoring corrupted underwater images remains a challenging task due to various factors affecting image quality. Therefore, a variety of underwater image restoration techniques (Fayaz et al., 2021; Wang et al., 2022b) have been proposed to enhance image detail information and increase image clarity. Underwater image restoration has become a hot research topic in the field of underwater image processing, and many significant results have been achieved. Most of the early traditional underwater image restoration methods were designed based on prior knowledge and physical models. Trucco & Olmos-Antillon (2006) first proposed a self-tuning image restoration filter based on the simplified Jaffe-McGlamery underwater imaging model, which optimized the local contrast quality judgment function to estimate the parameter values in the filter. Fan et al. (2010) proposed a new point spread function (PSF) and modulation transfer function (MFT) to address the underwater image restoration task. After the image has been filtered using the arithmetic mean filtering approach, the iterative blind deconvolution method is used to acquire the initial ideal value of the PSF of the corrupted image. He, Sun & Tang (2011) introduced a dark channel approach for recovering hazy images and the method is also extended to restore underwater images. In order to offer an R channel restoration approach for underwater image processing that significantly enhances color correction and sharpness, Adrian et al. (2015) created a connection between underwater image attenuation and wavelength. In order to restore underwater images and boost contrast and visibility, Cheng, Sung & Chang (2015) combined the R channel prior with the PSF’s physical characteristics. Zhang et al. (2017b) processed underwater color images using the multi-scale Retinex (MSR) method, and they created a multi-scale Retinex algorithm that greatly improves the images’ aesthetic impact. To boost detailed information and enhance the quality of underwater images, Sharanya & Ramesh (2013) used histogram equalization, a noise reduction filter, and a second-order directional derivative-based interpolation approach. The above methods show that, while they can slightly improve color, sharpen edges, and reduce image blur, the performance depends much on the amount of data and it is very hard to tune optimized parameters, and they are only suitable for images with minor degradation factors.

Due to their potent capacity for self-learning, deep learning, and neural networks have become increasingly important in image processing. Some researchers have used them for natural image processing (Dong et al., 2014; Kim, Lee & Lee, 2016; Zhang et al., 2017a; Liu et al., 2018) and produced outstanding outcomes. The deep learning-based approach has grown in popularity for underwater image restoration as a result of complicated underwater images. For instance, Li et al. (2017) trained an end-to-end network using a large number of underwater images and their depth data. The network can roughly estimate the depth of an underwater scene, restore the image, and improve visual effects. Fabbri, Islam & Sattar (2018) came up with the idea of using a UGAN network for underwater image restoration. They initially used the CycleGAN network (Zhu et al., 2017) to transform high-quality underwater images into low-quality images before training the network to improve the overall quality of underwater visual sceneries. In order to develop an underwater image enhancement network that has a respectable effect on underwater images, Li et al. (2019) suggested the WaterNet network. The network would involve the creation of a benchmark for underwater images as well as an underwater image quality improvement network. Islam, Xia & Sattar (2020) proposed the FUnIE-GAN network by varying the target function in various ways to repair underwater images by focusing on the image content, color, and texture information, and then designing the shallow network structure. However, there exists a limitation on generalization ability and the network itself cannot learn all the properties in the image. At present, Transformer has replaced the dominant architecture of CNN in the fields of natural language processing and computer vision. Because it is designed as a Transformer Block with a special multi-head attention mechanism to model the global dependency of the input image block sequence, it is widely applied to low-level visual task such as image restoration (Wang et al., 2021b; Wang, Pan & Tang, 2023; Huynh-Thu & Ghanbari, 2008).

Most of the above-mentioned methods extract single data features for processing, and therefore, produce the restoration effect. Deep learning-based algorithms have improved denoising performance and color correction excellently, but the ability to learn based on data properties is still limited.

Focusing on the above drawback, in this study, we propose a new underwater image restoration method—a hybrid attention network with a combination of CNN and Transformer. Convolutional neural networks are used for extracting local features. Therefore, the proposed method successfully combines the characteristics of the Transformer and CNN to fully learn global consistency data for future improvement. In addition, the network framework utilizes the hybrid attention mechanism, which combines self-attention, channel attention, and supervised attention, to successfully enhance the visual quality of the image by simultaneously removing noise and restoring image color. The network has a substantial impact on overcoming the shortcomings of conventional algorithms and the above-mentioned deep learning techniques in extracting single data features and can more effectively learn and explore various types of data features to further enhance image quality.

The contributions of this work can be summarized as follows:

• Proposing a hybrid attention network that combines CNN and Transformer. Through the feature extraction by CNN and Transformer Block, these features are utilized to learn global information and then fed into the network. So, the local and global features of underwater images are effectively fused.

• The proposed network framework incorporates a hybrid attention mechanism: Self-Attention (S-A) for modeling the global dependency relationships of the input image block sequence and extracting global feature information; channel attention for extracting inter-channel correlation; supervised attention for transmitting features between different parts, transferring the learned local features, and fusing them.

• Extensive experiment results demonstrate that the proposed method outperforms some baseline underwater image restoration techniques in terms of numerical evaluation and visual effects.

Related Works

Underwater image restoration methods based on local feature extraction

In recent years, with the continuous development of deep learning technology, underwater image restoration methods based on convolutional neural networks (CNN) and generative adversarial network (GAN) (Fabbri, Islam & Sattar, 2018) have become a research hotspot. They achieve better results in underwater image restoration tasks through local feature extraction using convolutional layers and subsequent enhancement. As a classic neural network, GAN can be directly trained in the domain of transformation between underwater images and clear images. To correct and enhance the color of underwater images while maintaining the edge structure information of the originals, Li, Bai & Niu (2023) suggested an enhanced water CycleGAN technique based on the edge structure similarity loss function. However, it may produce some artifacts. WaterGAN (Li et al., 2017) was proposed based on this, which uses a weakly supervised style transfer to generate a dataset of underwater blurry images, and then uses a correction network to restore unmarked real underwater images. Supervised learning using the GAN-generated dataset can make the trained model more adaptable to underwater scenes. The EUVP underwater dataset and the FUnIE-GAN network were generated by using the same technique (Islam, Xia & Sattar, 2020), which found that a combination of supervised and unsupervised techniques yielded good performance in restoration. However, the performance of contrast enhancement is limited due to the use of labeled images in supervised learning. Chen et al. (2017) used GAN to create a large number of underwater images and trained the model to improve model accuracy. Cao, Peng & Cosman (2018) proposed underwater image restoration with neural network topologies that measure background light and scene transmission rate. The input image is decomposed into many sub-bands using the discrete wavelet transform in UIE-WD. The algorithm includes a multi-color space fusion network and a detail enhancement network, which restore underwater images with rich high-frequency information in the frequency domain. Naik, Swarnakar & Mittal (2021) proposed Shallow-UWNet—a lightweight convolutional neural network structure, which concatenates the input image with the output of each residual block through skip connections. Considering the diverse lighting conditions in underwater environments, Liu et al. (2022) proposed LANet—an underwater image enhancement network based on attention mechanisms and adaptive learning that adaptively learns important feature information by implicitly perceiving lighting features.

Underwater image restoration methods based on global feature extraction

In order to accomplish the capability of global modeling, Swin Transformer (Liu et al., 2021) was suggested as a hierarchical Transformer structure that calculates features through a moving window. Uformer-B (Wang et al., 2021b) adapted the network encoding-decoding structure to Transformer with the ability to restore underwater images effectively. Peng, Zhu & Bian (2023) proposed a U-shape Transformer for underwater image enhancement by integrating the channel multi-scale feature fusion transformation (CMSFT) module and the spatial global feature modeling transformation (SGFMT) module to improve the network’s attention for color channels and severely attenuated spatial regions. Using Transformer for deep feature extraction and color correction modules to enhance image contrast and create enhanced underwater images, Zhang et al. (2023) proposed a two-stage network based on Transformer for single underwater image enhancement. It is challenging to be able to achieve both the low computational and low latency requirements for real applications due to the nature of the Transformer.

Attention mechanism

The attention mechanism is a technique used in deep learning that allows a model to selectively focus on certain parts of its input when making predictions or generating output. In computer vision, attention mechanisms are also used in various models to improve their performance on tasks such as image restoration and object detection. Currently, there are two common attention mechanisms, channel attention (Wang et al., 2021a; Wang et al., 2023) and self-attention (Wang et al., 2022a). The channel attention mechanism selectively emphasizes or suppresses certain channels of the feature map in the neural network according to different channels of attention. For example, Zhang et al. (2018) embed the channel attention mechanism into the residual block for super-resolution network RCAN, which greatly improves the performance of the network. Zhang et al. (2019) proposed a residual non-local attention network for high-quality image restoration. The self-attention mechanism aims to apply self-attention to feature maps to selectively focus on different spatial locations in the input image, and is often used in models such as non-local neural networks (Wang et al., 2018) and Transformer (Vaswani et al., 2017), etc.

Method

The objective function of the underwater imaging model

The majority of underwater imaging is made up of the direct component, the forward and backward scattered components, and the attenuated light that the camera receives. Underwater image model is often created using the Jaffe-MeGlamery (Trucco & Olmos-Antillon, 2006): (1) Ix=Jxtx+B1−tx+ηx

where Ix indicates the degraded image, J(x) denotes the clear image, J(x)t(x) means the direct component, B(1 − t(x))denotes the background scattering component, B denotes the underwater ambient light, and t(x) denotes the scene light transmittance, respectively. Another factor that commonly arises with underwater images is noise, which is denoted by a parameter ηx, and usually additive noise.

To map the degraded image to the clean image with a deep learning approach, we typically train a neural network. Consequently, the map F:I(x) → J(x) can be defined as an objective function of a minimization problem in the following form: (2) arg minJx12||Jx−FIx||22

The above problem is equivalent to the following problem: (3) minLFNety;θ,x

where FNet is the network model that we suggested, L is the loss function, y is the network’s input of a degraded image, x is the clean image, and θ is the learning parameter.

Hybrid attention network combining CNN and Transformer

We propose a hybrid attention network (MA-CTNet) incorporating CNN and Transformer for underwater image restoration. Figure 1 presents an overall of the network architecture. The global feature extraction component, the local feature extraction component, and the feature fusion strengthening component make up the entirety of the network structure. Two residual groups (RGS) with a convolutional layer, residual channel attention block (RCAB), and channel attention (CA) mechanism are used to extract local features and enhance shallow features of input images. As local feature augmentation cannot rectify the overall color bias of damaged underwater images, the Transformer Blocks structure is integrated into the global feature extraction phase to replicate the global dependency of input image block sequences utilizing Self-Attention (S-A). Then, it extracts the details of the global feature. The improved local and global features are then combined together. The Supervised Attention (SA) mechanism is used to achieve feature transmission prior to the reinforcement phase. The SA module uses the real image as the supervision constraint, transfers locally learned features, fuses them with globally learned features, and inputs them into the reinforcement phase to further mine the image details, and then bolsters and reestablishes the deep features.

Figure 1 A hybrid attention network framework with CNN and Transformer.

Local feature extraction

Figure 1 illustrates the components of the local feature extraction, which include a convolution layer (Conv), two residual groups (RGS), and skip connections. Two residual channel attention blocks (RCABs) with CA mechanisms are present in each residual group. CA focuses on significant channel characteristics and utilizes excellent recognition ability to extract and enhance deep features and improve deep network features during restoring texture and details (Zhang et al., 2018). Utilizing the interdependence between channels through skip connections, the useful local feature information can be recovered even in the deeper network by incorporating CA in the local feature extraction portion. As illustrated in Fig. 2, CA is made up of an average pooling layer, two convolutional layers, ReLU and Sigmoid activation functions, and residual connection. To create the compressed feature map, a pooling layer first down-samples the input feature. Dimension reduction, two convolutions, and the ReLU activation function are all used to learn the nonlinear interaction between several channels. Finally, the output of Sigmoid function is element-wise multiplied by its input to provide an output of the same magnitude when it is activated.

Figure 2 Channel attention.

Global feature extraction

It can be challenging to recover the entire color of an underwater image using only local features. Deriving from the core of the Transformer mechanism of Self-Attention, we can model the interdependence of input and output image sequences without regard to their closeness in the sequence. Therefore, the global feature extraction achieved a better performance in the underwater image restoration. Thus, as shown in Fig. 1, the part for extracting global features consists of two Transformer Blocks with Self-Attention and one Patch embed (3x3 Conv layer). Through the convolution of the input images, patch embedding can provide low-level feature embedding. Then, in order to achieve global recovery, the underlying characteristics are put into Transformer Blocks to extract global features. Figure 3 depicts the composition of the transformer blocks, which are made up of Multi-DConv Head Transposed Attention (MDTA) with Self-Attention, and feed forward network (FFN) blocks. In more detail, after the underlying feature has passed through LN, the pixel-level cross-channel context is aggregated using 1 × 1 convolution, and the channel-level spatial context is then encoded using 3 × 3 depth-level convolution to produce the query (Q), key (K) and value (V) projections, denoted Q=WdQWpQX, K=WdKWpKX, V=WdVWpVX. Here, Wp⋅ is the 1 × 1 point-wise convolution, and Wd⋅ is the 3 × 3 depth-wise convolution. S-A is calculated on channels to reduce computational complexity: X ˆ=WpSAQ,K,V+X

(4) SAQ,K,V=V⋅SoftmaxK⋅Q/α

Figure 3 Transformer blocks.

Here, α is a learnable scaling parameter used to regulate the size of the dot product and before applying the softmax function, X and X ˆ are the input and output feature maps. Additionally, Wp is the weight of the 1 × 1 point-by-point convolution. In order to acquire the required number of channels, a 1 × 1 convolution layer is employed after the normalized Softmax coefficients are converted back to feature V. The channels are separated into ‘heads’, and several attention graphs are learned concurrently for each head.

FFN uses two 1 × 1 convolutions: one to extend the feature channel and the remaining one to reduce the channel back to the original input dimension; and nonlinear GeLU activation function in the hidden layer to effectively recover from adjacent pixel positions. FFN is expressed as follows: (5) X ˆ=Wp0ϕWd1Wp1LNX⨀Wd2Wp2LNX+X

where the operator ⊙depicts element-wise multiplication, ϕ is nonlinear GeLU activation function, LN(⋅) isthe normalization layer, Wp⋅ is the 1 × 1 point-wise convolution, and Wd⋅ is the 3 × 3 depth-wise convolution.

Feature fusion and enhancement

The supervised attention SA mechanism is integrated before the feature reinforcement part. On the one hand, focusing on using the real image as the supervision constraint in order to restore the real image; on the other hand, supervised attention SA can generate attention feature maps in order to suppress features with less information at the moment and only pass useful features to the next stage. Local and global features can be successfully fused to strengthen deep features. The enhanced component has the same structure as the local feature extraction part, and residuals are employed for deep enhancement. Figure 4 depicts the SA mechanism of supervision attention. The recovered local features are first convolution by a 3 × 3 convolution layer and superimposed on the deteriorated image to generate the intermediate output and calculate its Charbonnier loss with the ground-truth image GT. One 1 × 1 convolution layer and Sigmoid activation function are used to activate the intermediate output, which is then element-wise multiplied and overlaid with the local feature to generate the attention feature map of the output as follows:

Figure 4 Supervised attention.

O1 = Conv(xin + X) (6) xout=SigmoidConvO1⊙Convxin+xin

where the operator ⊙ represents the element-wise multiplication, Conv(⋅) represents the convolution operator.

Loss function

The hybrid attention network structure paired with CNN and Transformer has the following total loss function form: (7) L=L1O,GT+LCharbonnierO1,GT

Among them, the loss function contains two terms: loss L1 and Charbonnier loss LCharbonnier, respectively. GT represents the ground-truth image, O is the image recovered by the overall proposed network, and O1 represents the output image obtained by the supervision and attention modules. The constraint of Charbonnier loss  (Charbonnier et al., 1994) is to extract useful features by using supervised learning, and the Charbonnier loss function is defined as follows: (8) LCharbonnier=||O1−GT||2+ɛ2.

To prevent the training network from vanishing gradient, we have added a numerical stabilization constant ɛ and usually set it to 0.001.

Experimental Results

Experimental setup

We implemented the performance of our method on two datasets, EUVP (Adrian et al., 2015) and UIEB (Jaffe, 2015). The Underwater Dataset EUVP consists of 5,550 paired training images, we chose 5,022 images to train and 528 images to test during the experiment. For UIEB, we randomly selected 800 images in the UIEB dataset as the training set and the remaining 90 images as the testing set.

During network training, the batch_size and patch_size parameters are set to 16 and 48, respectively. The initial learning rate is set at 10−4, and it is reduced to half every 2 × 105 back-propagation iterations. The Adam optimizer was used, and network convergence occurred when the iteration epoch reached 300. Simultaneously, the best model was chosen for testing. Furthermore, Gaussian noise was reproduced on the original data set for training and testing to validate the denoising impact.

As the comparative methods, one traditional method DCP (He, Sun & Tang, 2011), four other known deep learning image restoration methods, namely CycleGAN (Zhu et al., 2017), FUnIE-GAN (Islam, Xia & Sattar, 2020), Shallow-UWnet (Naik, Swarnakar & Mittal, 2021), and RTFAN (Fu et al., 2021), are employed in this research. To ensure fairness and dependability, all methods use the same training and test sets as described in this work. We implemented the experiments on PyTorch framework with the NVIDIA GeForce RTX 3090 GPU system.

Quantitative analysis

To evaluate the performance of the methods, we use two image quality assessment metrics: peak signal-to-noise ratio (PSNR) (Huynh-Thu & Ghanbari, 2008) and structural similarity index measure (SSIM) (Kotevski & Mitrevski, 2010). PSNR characterized how close a processed version of an image is to the ground truth, by calculating the mean square error (MSE) between the original image and the processed image: (9) PSNRx,y=10log102552MSEx,y

where, x and y are the reference image and the image to be evaluated respectively. MSE() represents the mean square error operation, and the larger the PSNR value, the better the image quality. In the meanwhile, SSIM takes into account the brightness, contrast, and structure of two images, therefore, closer to human perception as Formula (10). (10) SSIMx,y=2μxμy+c12σxy+c2μx2+μy2+c1σx2+σy2+c2

where, μx and μy are the mean, σx2 and σy2 are the variance, σxydenotes the covariance, c1 and c2 are the constant that’s used to maintain stability.

In addition, UCIQE (Yang & Sowmya, 2015) is used to quantitatively evaluate underwater images’ non-homogeneous color shift, blurring, and low contrast. It is a linear combination of color intensity, saturation, and contrast as follows: (11) UCIQE=c1σc+c2conl+c3μs

where σ is the standard deviation of the image, and it can represent the average of saturation.

All the methods are tested on EUVP test set and test sets with different Gaussian noises with zero mean and noise variance σ2 with values 5, 10, and 15, respectively. Table 1 presents the quantitative results of the proposed method and various underwater image restoration algorithms.

Table 1 The PSNR and SSIM scores of the proposed method and various underwater image restoration algorithms (bold values are the best).

Algorithm	Running times(s)	Evaluating indicator	Dataset EUVP + Gaussian noise with σ 2	
			5	10	15	
CycleGAN	179	PSNR/SSIM	16.35/0.7773	16.27/0.7295	16.26/0.6858	
FUnIE-GAN	40	PSNR/SSIM	20.65/0.8632	20.23/0.8076	20.24/0.7764	
Shallow-UWnet	321	PSNR/SSIM	20.67/0.8663	20.57/0.8309	20.03/0.7822	
RTFAN	80	PSNR/SSIM	22.36/0.8936	22.19/0.8708	21.97/0.8498	
Ours	109	PSNR/SSIM	22.49/0.8970	22.27/0.8752	22.03/0.8535	

It can be shown that the proposed method has higher performance in PSNR and SSIM scores, which can be increased by 0.13 dB in the best scenario. Furthermore, we show the exponential curve of the data during training and use it to quantify and analyze convergence. Figure 5 (left) depicts the trend of the loss curve. Figure 5 (right) depicts the trend of the PSNR curve. From Fig. 5 (left), it can be seen that the loss function of our method gradually tends to a downward trend, and when the epochs is 300, it gradually becomes flat. Figure 5 (right) shows the trend of increasing PSNR of the validation set as the number of iterations increases.

Figure 5 The loss function curve (left) and the PSNR peak of the verification set during the training process (right) of the proposed network.

To fully validate the effectiveness of our method for different degraded images, we quantitatively compare with a traditional method and the four competing deep learning-based methods. we computed the PSNR, SSIM, and UCIQE results on the UIEB dataset in Table 2.

Table 2 The quantitative results of the proposed method and various underwater image restoration algorithms (bold values are the best).

Algorithm	Running times(s)	Dataset UIEB	
		PSNR	SSIM	UCIQE	
DCP	13	15.84	0.6152	0.5743	
CycleGAN	33	18.78	0.6494	0.4995	
FUnIE-GAN	3	17.76	0.6127	0.5673	
Shallow-UWnet	54	18.09	0.6683	0.5126	
RTFAN	74	20.08	0.7283	0.5844	
Ours	96s	20.85	0.7601	0.5874	

From Table 2, it can be seen that it is the best in PSNR, SSIM, and UCIQE metrics, although our method is slightly inferior in running time. Therefore, the proposed method achieves the best quantitative evaluation results on two datasets.

Qualitative analysis

When the noise variance is 5, five different test images are selected on the EUVP test set for comparison, and the restored visual results are shown in Fig. 6.

Figure 6 Results of restoration of certain underwater images with noise variance 5.

(A) Input, (B) CycleGAN, (C) FUnIE-GAN, (D) Shallow-UWnet, (E) RTFAN, (F) ours, and (G) Ground Truth.

As shown in Fig. 6, unlike existing convolutional neural network methods that locally enhance image characteristics such as unevenly distributed color areas, the proposed method focuses on overall image consistency restoration, particularly for some vividly colored images. The proposed method suits to resolve the issue of image color bias and restore image features. The proposed method suits to resolve the issue of image color bias and restore image features. In addition, the proposed method eases image color bias (line 1 and line 5) and recovers more background details (lines 2–4) than other underwater image restoration methods.

To verify that our method has a good recovery effect on different types of data, we show the visualization results of the UIEB test set as shown in Fig. 7. In particular, we selected and processed degraded images containing green to ensure restoration. Due to space constraints, we choose the traditional enhancement algorithm DCP and one deep learning methods for underwater image restoration for comparison.

Figure 7 Results of restoration of certain underwater images on UIEB datasets.

It can be seen from Fig. 7 that when the degraded underwater image appears green, the restoration results of our method have obvious visual advantages compared with other underwater image restoration methods. For example, DCP cannot handle this kind of image with a severe color cast. RTFAN will have an overall yellowish effect, and the restoration effect is not outstanding.

Ablation experiment

To fully verify the effectiveness of the proposed method in the color cast, we conduct an ablation experiment on Transformer Blocks and Channel Attention, and the quantitative and qualitative results are shown in Table 3 and Fig. 8, respectively.

Table 3 Comparison of quantitative results of ablation experiment (bold numbers are the best).

Dataset	Noise variance σ 2	Module	PSNR	SSIM	
EUVP	10	w/o Transformer Blocks	22.14	0.8744	
EUVP	10	w/o CA	22.19	0.8751	
EUVP	10	w/ Transformer Blocks, CA	22.27	0.8752	

Figure 8 Visual effects of the ablation experiment.

Table 3 shows the benefits of adding Transformer Blocks (TBs) to further showcase the ability in fusing local and global features. At the same time, CA is tested for improving the color cast of the image as shown in Figs. 8B and 8D.

Comparing the results with/without using Transformer Blocks and CA as in Fig. 8, Both considerably improve the restoration results of overall color and details.

In addition, we also experimented with the number of Transformer Blocks in our network, and we tested the PSNR and SSIM for the cases of Transformer Blocks 1, 2, 3, and 4 on the EUVP dataset as shown in Table 4.

Table 4 Ablation experiment of different Transformer Blocks (bold numbers are the best).

Transformer Blocks	PSNR	SSIM	Training Time(h)	
1	22.24	0.87530	15.28	
2	22.27	0.8752	19.43	
3	22.25	0.87518	23.66	
4	22.25	0.87614	25.06	

Hence, according to the above quantitative results and consideration of training time, we still choose two Transformer Blocks.

Conclusion

We have proposed a novel hybrid attention network for underwater image restoration that combines CNN and Transformer. In the network, convolutional neural networks are mainly concerned with extracting local features, Transformer Blocks focus on global characteristics, and then fuse local and global information. The proposed network can resolve the major color bias issue of the underwater map. In addition, channel attention and supervised attention are added to mine the association between channels and features and refine the deep features while taking into account a range of degradation variables. Experimental results verified the efficacy of the proposed method in restoring image consistency, in general, and vividly colored regions, in particular, and improving image details. Although the hybrid attention network performs well in processing underwater images, the performance of the hybrid attention network may be limited for some very complex images, such as images containing a lot of noise and interference. Moreover, it is very difficult to obtain high-quality training data in an underwater environment, and underwater image restoration needs to deal with many different types of data, such as images, sounds, sensor data, etc. Therefore, how to integrate multiple data to improve the performance of hybrid attention networks is one of our future research directions.

Supplemental Information

Supplemental Information 1 Source code with implementation on Python

Click here for additional data file.

Additional Information and Declarations

Competing Interests

Author Contributions

Data Availability

The authors declare there are no competing interests.

Zhan Jiao conceived and designed the experiments, performed the experiments, analyzed the data, performed the computation work, prepared figures and/or tables, authored or reviewed drafts of the article, and approved the final draft.

Ruizi Wang conceived and designed the experiments, performed the experiments, analyzed the data, performed the computation work, prepared figures and/or tables, authored or reviewed drafts of the article, and approved the final draft.

Xiangyi Zhang conceived and designed the experiments, performed the experiments, analyzed the data, performed the computation work, authored or reviewed drafts of the article, and approved the final draft.

Bo Fu analyzed the data, authored or reviewed drafts of the article, and approved the final draft.

Dang Ngoc Hoang Thanh analyzed the data, authored or reviewed drafts of the article, and approved the final draft.

The following information was supplied regarding data availability:

The EUVP dataset is available at https://irvlab.cs.umn.edu/resources/euvp-dataset.

The UIEB (underwater image enhancement benchmark) data is available at: https://opendatalab.com/UIEB/.

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
