# Peer review of "A hybrid attention network with convolutional neural network and transformer for underwater image restoration"

_PeerJ Computer Science, doi:10.7717/peerj-cs.1559_

## Round 0.1 · original submission · Major Revisions

Dear Authors,

Your paper has been reviewed. It needs major revisions before being accepted for publication in this journal. More precisely, you have to conduct additional experiments to improve the validity of your findings. Furthermore, you have to revise the manuscript style and language extensively.

Reviewer 1 ·

Basic reporting

Reviewer’s Report on the manuscript entitled:

A hybrid attention network with convolutional neural network and transformer for underwater image restoration


The authors proposed a hybrid attention network for underwater image restoration that combines CNN and Transformer. They tested and compared their model using EUVP dataset. The method and results are interesting, but the presentation and literature review can be improved. Please see below my comments for further improvement.

Experimental design

Equation (4). Please define parameters Q and K.

Table 1. Please add the computational time for each method on the same PC you used.

Line 317. Please describe these metrics mathematically.

Figure 6. What is panel (e)? You forgot to mention it in the caption. Also, please add labels (a), (b), …. In the figure.

Validity of the findings

A discussion section can be added to discuss the method and results and comparisons with other methods. Areas of improvement, other applications, etc. can also be added to the discussion part. I also suggest showing additional statistical metrics, e.g., overall accuracy, confusion matrices, RMSE.

Additional comments

Please also discuss the following recent article: https://doi.org/10.3390/jmse11040787

In the discussion section, please also discuss deep transfer learning architectures, such as VGG and ResNET that can be used for image enhancing, denoising, and classification.

Please mention the limitations of your method and future direction in the conclusion section.

Thank you!

·

Basic reporting

The Authors proposed in this paper “A hybrid attention network with convolutional neural network and transformer for underwater image restoration”. I commend them for their extensive work; from introduction to literature, and from methodology to experimental results, and the conclusion. In addition, the manuscript is clearly written in professional, unambiguous language. However, below are my few constructive comments.
The abstract, though was well written; there is a need to include the results and the implications.

Experimental design

The Experimental results section should contain section for Discussion where the mentioned existing methods can be extensively compared and contrasted with Authors’ proposed methods.
More Ablation experiment is needed.

Validity of the findings

No comment

Reviewer 3 ·

Basic reporting

The authors propose a hybrid attention network combining CNN and Transformer for underwater image enhancement. However, there are still some problems.
1) In line 35, “Real underwater images”, what does "real" mean here?Does it mean “clear”? If you mean “clear”, there's a problem, because in fact you cannot know what Real(clear) underwater images are.
2) The innovation of this article is about Transformer, but there is almost no description of Transformer in the introduction section.
3) Why are "two" Transformer blocks used? Have you tried any other number of Transformer blocks?
4) Introduction and Related Works both describe the current research status of traditional methods and deep learning methods, with similar content.

Experimental design

Additional experiments are needed.
1) Are the results of qualitative and quantitative experiments using 528 images as the test set lacking persuasiveness?
2) The comparison algorithm only uses four deep learning methods, please consider whether it is representative and whether it needs to be compared with traditional restoration methods?
3) In qualitative analysis, most of the displayed degraded images are blue, but the degree of degradation in underwater images varies. How effective is it when dealing with green or other types of degraded images?
4) Lack of ablation research on core components.

Validity of the findings

1) The effectiveness of the proposed method in the color cast has not been fully verified by relevant experiments.
2) Line 108, contributions of this work state "Extensive tests are carried out on several underwater image datasets", but it is stated in line 25 of the abstract and line 301 of the experimental section that the experiment was only conducted on the EUVP dataset.
3) The description of the experimental results in Figure 6 and Figure 7 is not sufficient.

Additional comments

The article requires extensive language revisions, and the worst part may be the introduction.
1) There are the improper matchings or unclear expressions, such as "The topic of restoring corrupted underwater images is difficult and poorly posed." in line 41; the title of 2.2 in line 133 is "Underwater image restoration methods based on CNN", but the content mentions GAN; line 198, "CNN releases on local feature extraction..."; line 334, "Figure 5 (left) focuses on the history of the loss curve, etc.
2) There are details problems, such as inconsistent spacing between lines containing formulas and variables; the PSNR indicator in line 317 lacks citation; the format of reference [39] in line 473, etc.

---

## Round 0.2 · accepted · Accept

Dear Authors:

Your manuscript has been accepted for publication in PeerJ Computer Science. The comments of the reviewers who evaluated your manuscript are included at the foot of this letter. I ask that you make minor changes to your manuscript based on those comments, before uploading the final files.

Reviewer 1 ·

Basic reporting

Dear authors,

Thank you for addressing my comments satisfactorily and improving your manuscript. I have a few minor suggestions.

In response to my Q6, you wrote "The residual block in ResNet is also used in our network structure". However, I do not see that you mention it in the Discussion in the light of the article I suggested. Please note that early stopping and gradient clipping could potentially enhance the learning process while avoiding over-fitting. These could at least be mentioned in the discussion section.

Regards,

Experimental design

no comment

Validity of the findings

no comment

Additional comments

Please carefully proofread the manuscript and correct the typos/punctuation issues.

Reviewer 3 ·

Basic reporting

The authors have carefully revised the manuscript based on the reviewers’ comments. Most of my comments were properly addressed. I do not have any more comments.

Experimental design

The experiment design is adequate to demonstrate technical feasibility.

Validity of the findings

The evaluation metrics can prove the validity of the findings.

Additional comments

I have no further comments.